# Prevalence of violence in a clinical sample of adolescent patients visiting a child and adolescent psychiatric outpatient clinic in Nepal

Rampukar Sah[1,2,3]*, Per Håkan Brøndbo[1], Jasmine Ma[2], Ketil Lenert Hansen[4], Narmada Devkota[2,5], Bjørn Helge Handegård[4], Anne Cecilie Javo[6]

1 Department of Psychology, Faculty of Health Sciences, UiT-The Arctic University of Norway, Tromsø, Norway, 2 CWIN-Nepal CAPMH Research and Outreach Center, Kathmandu, Nepal, 3 Child & Adolescent Psychiatry Unit, Kanti Children's Hospital, Kathmandu, Nepal, 4 Regional Centre for Child and Youth Mental Health and Child Welfare—North, Faculty of Health Sciences, UiT-The Arctic University of Norway, Tromsø, Norway, 5 Adolescent Mental Health Unit, Mental Hospital, Patan, Nepal, 6 SANKS—Sámi Klinihkka, Finnmark Hospital Trust, Karasjok, Norway

* rampukarsah@yahoo.com

## Abstract

### Background

Child violence is a global concern affecting the well-being and development of children and adolescents worldwide. Despite the obvious need, few studies on child violence have been conducted in clinical samples, especially in low- and middle-income countries.

### Objective

The aim of this study was to examine the prevalence of different types of violence in past-year among adolescent psychiatric patients in Nepal.

### Participants and setting

The participants were 810 adolescents aged 11–15, 392 boys and 418 girls, who visited a child- and adolescent psychiatric outpatient unit in Kathmandu during a 12-month period.

### Methods

We used a descriptive, quantitative, cross-sectional design. Data was collected with screening instruments completed by the adolescents themselves. Prevalence rates and range of occurrence of various forms of child violence were computed for both genders. Gender comparisons were conducted using Pearson chi-square tests. Adolescents rated the occurrence in the "rarely", "sometimes", "often" or "frequently" categories. Associations between the different forms were examined by Spearman's correlation test.

**Data availability statement:** All relevant data are within the paper and its Supporting Information files.

**Funding:** This study led by Rampukar Sah is funded by the Norwegian Partnership Program for Global Academic Cooperation (NORPART) 2018/10039 project ("Collaboration in Higher Education in Mental Health between Nepal and Norway") and Child Workers In Nepal (CWIN)-Nepal. The NORPART project funded for the travel and accommodation for the PhD courses. The CWIN-Nepal funded for the research and the salary of the principal investigator. URL: 1. NORPART: https://diku.no/en/programmes/norpartnorwegian- partnership-programme-for-globalacademic-cooperation 2. CWIN-Nepal: https://www.cwin.org.np/. The funders had no role in study design, data collection and analysis, decision to publish, or preparation of the manuscript.

**Competing interests:** No conflict of interest.

## Results

In this study 88% of adolescents had experienced some forms of violence, girls reporting higher prevalence than boys last year. Emotional abuse was the most common. Neglect was reported by 25% of the adolescents, and domestic violence by 40%. Sixty percent of the adolescents had experienced peer aggression. Nearly 75% of the adolescents had experienced polyvictimization and it was higher in girls than boys. Significant correlations were found between several forms of violence, indicating compounded risks.

## Conclusions

The study demonstrates high prevalence of multiple forms of violence among adolescent psychiatric patients, calling for increased awareness of child violence in young patients admitted to mental health institutions in Nepal.

## Introduction

Violence against children is a pervasive global issue, affecting billions of young lives every day [1]. According to the World Health Organization (WHO), exposure to violence in childhood is a serious problem worldwide [1], with far-reaching impact for both short-term and long-term mental health [2]. The United Nations Convention on the Rights of the Child (UNCRC) defines violence against children as all forms of physical or mental violence, injury or abuse, neglect or negligent treatment, maltreatment, or exploitation, including sexual abuse, encompassing violence within and outside the home and involving both adults and children [3]. Knowledge about the magnitude of child maltreatment and types of violence is critical as it is strongly linked to the development of mental disorders later in life. Research reported that about 50% of adults with psychiatric disorders have experienced childhood maltreatment such as child abuse and neglect [4–7]. Physical abuse is characterized by the intentional use of force by a parent or caregiver against a child, causing physical harm [8] while emotional abuse involves any intentional verbal or symbolic acts by a parent or caregiver that pose a significant risk of emotional harm [9]. Child sexual abuse, involves the participation of a child in sexual activity that he or she does not fully understand, cannot give informed consent to, or is developmentally unprepared for [10]. This stark figure (50%) demands attention as exposure to abuse during key developmental periods heightens vulnerability to a range of psychiatric conditions, including mood disorders, anxiety, substance use, antisocial behavior, and psychosis [11–15]. Moreover, environmental factors like family support, domestic violence, and cultural context can either worsen or mitigate the effects of childhood maltreatment [16,17]. Children from supportive environments are more likely to develop resilience, thereby reducing their risk of later psychiatric issues [16,17]. On the other hand, parental neglect is regarded as the most common form of child maltreatment and is perceived as being at the root of all forms of child maltreatment because an abusive

parent is most often simultaneously neglecting the child, not fulfilling his/her parental role and responsibilities [18]. Further, there is significant evidence on the link between domestic violence and violence against children in recent data. Studies have shown that violence against women and violence against children intersect in different ways [19,20]. Witnessing domestic violence is considered in and of itself a form of violence against children, having significant consequences for them. Finally, violence against children also includes peer victimization. Peer victimization is a problem of rising concern due to the high prevalence and negative consequences for the mental health of adolescents [21,22]. A former international meta-analysis encompassing data from 80 studies showed that 36% of all adolescents felt victimized by their peers [23].

The present study encompassed all the above-mentioned forms of child abuse, child neglect, domestic violence, and peer aggression, and used as its target population a clinical sample of adolescents in Nepal – a less investigated low- and middle-income country (LMIC). Globally, the prevalence of violence against children is at its peak. A recent systematic review which provided data for 96 countries on past-year prevalences of violence against children, estimated that a minimum of 50% or more of children in Asia, Africa, and Northern America experienced physical, sexual, emotional, or multiple types of abuse [24]. It found that globally, over half of all children – more than 1 billion children aged 2–17 years – experienced such violence [24]. However, violence estimates vary considerably among countries, and wide differences exist by violence type (physical, emotional, and/or sexual) and by gender [25]. Prevalence rates of violent disciplinary methods seem to be particularly high in South Asian and Sub-Saharan African countries [24,26]. Also, studies in the South Asian region show that witnessing violence, either between parents or between known adults, is frequently reported by children [26]. Further, studies done on the pattern of violence in LMICs indicate that children and adolescents in these countries may experience higher levels of polyvictimization (the co-occurrence of multiple violence forms) than in high-income countries (HICs) [27]. Research has reported that children who are exposed to one type of violence are at a significantly higher risk of experiencing other types, too [28]. This may have important clinical implications as polyvictimization has been associated with increased likelihood of mental health problems and have additive effects on mental health outcomes [29–31]. However, relatively little is known about polyvictimization and the different forms of violence against children in LMICs as we still lack local data from many of these countries, including Nepal. Hence, more research is required to address the knowledge gap, including possible gender differences in prevalence, associations between the different forms of violence, and the co-occurrence of multiple violent forms.

The existing literature suggests that the prevalence of all forms of child maltreatment will be higher in clinical samples than in the general population as violence against children is a risk factor for psychopathology and social dysfunction [18,32]. In LMICs, prevalence data on child violence in clinical psychiatric samples are limited to non-existing. Presumably, prevalence might be similar or even higher in these countries. In Nepal, no study of violence against children in clinical psychiatric samples has been done except for a recent descriptive study that examined social-demographic and clinical profiles of children and adolescents attending the psychiatric outpatient service of a tertiary level hospital in one region of Nepal. The study reported that 8% of new cases were related to child abuse [33]. However, the study did not fully explain how child abuse was defined and measured. Hence, it is difficult to assess its results. Till now, no extensive study examining the prevalence of self-reported child violence in a clinical sample has been conducted in Nepal.

## Aims of the study

The present study aimed to assess the prevalence and occurrence rates of the various forms of violence against children in Nepal, i.e., child abuse, child neglect, domestic violence, and peer aggression, as reported by adolescents aged 11–15 years who were visiting a child and adolescent psychiatric outpatient unit in Kathmandu during a 12-month period (2023/2024). We also aimed to assess the prevalence for boys and girls, as well as the prevalence of polyvictimization and gender difference for all forms of violence, along with polyvictimization. Further, we aimed to examine the correlation between the different forms of violence.

## Methods and materials

### Study site and population

Nepal is a country in the Southern subregion of Asia which is defined in both geographical and ethnic-cultural terms. The subregion also includes Bangladesh, Bhutan, Nepal, India, Pakistan, Sri Lanka, Maldives, and Afghanistan. Nepal is a multicultural country with 142 castes and ethnic groups of which the majority is Hindu [34]. The Nepali population, which reached 31 million in 2024, is a young population with about 35% (10.5 million) under working age (i.e., the age of 15) [35]. The country has poorly developed mental health services, and the accessibility of such services is still limited for the more remote areas due to lack of specialists, long distances, and poor communication facilities. Child and adolescent psychiatry services were only recently established and are in the process of being further developed. In 2015, the first child- and adolescent psychiatric outpatient unit (CAP-OPD) was established at Kanti Children's Hospital (KCH), Kathmandu, and has since then been gradually expanding. The present study used patients who visited the KCH CAP-OPD as its informants. This study was approved by the Kanti Children's Hospital Ethical Review Committee (Reference No: 621) and the Nepal Health Research Council (Reference No: 1227).

### Participants

The participants were adolescents aged 11–15 and their parents who had been admitted to the CAP-OPD in a 12-month period, from 15/01/2023 to 15/01/2024. From a total number of 1590 new patients in the period, 642 were excluded. Exclusion criteria were patient diagnoses such as intellectual developmental disorder (IDD), acute psychosis, or other conditions impairing cooperation. Of the remaining 948 patients who were invited to participate in the project, 138 refused to participate, mostly due to parents' busy schedule, resulting in a final sample of 810 patients and a response rate of 85%.

### Study design

The present study used a clinical sample, and was designed as a descriptive, quantitative, cross-sectional study.

### Procedure

The data were collected by two office secretaries with a bachelor's degree in education and social work, and a research assistant with a master's degree in counselling psychology. Before commencing data collection, they received intensive one-day training from the researcher (first author), which covered information about the aims of the project, the use of the different instruments, and how to inform and assist parents (in particular, illiterate parents) and adolescents in filling out the forms. Throughout their work, they were closely monitored by the researcher.

All participating adolescents and parents were informed, both orally and in written form, about the project. Informed, written consent was obtained from all participants. The adolescents were asked to fill out screening instruments about abuse, neglect, home environment, and peer victimization, while their parents were asked to fill out a socio-demographic background information form. The different forms were completed by each adolescent alone in a separate, quiet room, prior to a clinical consultation, and with trained secretaries available to answer any queries and to clarify possible misunderstandings. Subsequently, the clinicians responsible reviewed the filled-in forms that the participants brought with them to the consultation room. Data plotting was done manually throughout the collection period.

### Instruments

**Screening instruments of child maltreatment.** The screening form for child abuse that was filled out by the adolescents, asked about their experiences of the different forms: physical abuse, psychological abuse, and sexual abuse. An additional form asked about child neglect and home environment as well as peer victimization.

The five questions about the different forms of child abuse were taken from the *Pediatric Hurt-Insult-Threaten-Scream-Sex (PedHITSS)* screening tool [36]. This 5-item questionnaire was designed to detect and prompt provider investigation into child abuse in clinical settings and had been validated by comparing it to the Conflict Tactics Scale: Parent-Child Version (CTSPC) screening measure [36]. Responses are given on a 5-point Likert scale (0 = never, 1 = rarely, 2 = sometimes, 3 = fairly often, 4 = frequently). PedHITSS for parents contains the following questions: "During the last year, how often would you estimate that an immediate family member did each of the following to the child: (1) Physically hurt him/her; (2) Insult him/her or talk down to him/her; (3) Threaten him/her with physical harm; (4) Scream or curse at him/her; or (5) Force him/her to have sex [36]. The questions related to insults, threats and curse were categorized as "emotional abuse". The questionnaire for adolescents was modelled after the PedHITSS parent form. It has a similar format, only those questions were worded in the first person. We adapted the questionnaire by adding questions about the types of perpetrators. We asked and received permission to use the extended questionnaire by the owners of PedHITSS.

Questions about child neglect and home environment were posed in an additional questionnaire which also included a section with questions about peer victimization. The question about child neglect was: "During the last year, did you experience any of the following events at home?", with the following 6 statements (options): "You did not get enough to eat (went hungry) and/or drink (were thirsty)", "You had to wear clothes that were dirty, torn, or inappropriate for the season", "You were not taken care of when you were sick or injured", "You were hurt or injured because no adult was supervising", "You did not feel cared for", "You were made to feel unimportant".

The overall question about home environment/ domestic violence was: "During the last year, how have you experienced your home life?" with the following 6 questions: "Do you feel safe in your family?", "Has anyone in your home used alcohol and/or drugs and then behaved in a way that frightened you?", "Have you seen adults in your home shouting and screaming in a way that frightened you?", "Have you seen adults in your home hurt each other physically (e.g., hitting, slapping, or kicking)?", "Have you seen adults in your home use knives, guns, stick, rocks or other things to hurt or scare someone else inside home?", "Have you been mistreated or bullied by your brother(s) or sister(s) at home?" The adolescents could put more than one mark if several options fit.

The two questions about child neglect and home environment were taken from the *ISPCAN Child Abuse Screening Tool Children's Version (ICAST-C)* which has altogether 38 questions. The ICAST-C is a multi-national, multi-lingual, consensus-based survey instrument that has been tested in many countries [37]. As ISPCAN members, we got permission from the ISPCAN group to use the two ICAST-C questions for our study.

The questions about peer victimization were: (1) "During the last year, how often have you been hit or physically bullied by other children?" (2) "During the last year, how often have you been teased or harassed by other children? (for example: others have made fun of you, teased you, or humiliated you)." (3) "During the last year, how often have you been isolated or left out of things on purpose by other children?" (4) "During the last year, how often have you experienced cyberbullying? (meaning that another child or group of children used the internet to say bad things about you, humiliated you, or made fun of you)".

Responses were given on a 5-point Likert scale (0 = never, 1 = rarely, 2 = sometimes, 3 = fairly often, 4 = frequently) for each question. All questionnaires used in our study were translated and back translated into Nepali by clinicians and professional translators.

**Background Information Questionnaire.** The parents were asked to fill out a questionnaire asking about socio-demographic data. It provided information about gender, age, living area, geographical region, type of family, parental education, parental employment status, and family annual income, among others.

## Ethical considerations

Ethical clearance was granted from the local ethical board, the Institute Review Council (IRC) of Kanti Children's Hospital (KCH) and from the national ethical review board, the Nepal Health Research Council (NHRC). As part of the procedure

of collecting data, both written and oral information about the study were given to all participants, including information on confidentiality and the possibility to withdraw from the study. Both collection and storage of data were conducted according to the strict rules of the NHRC and IRC. All personally identifiable information was removed from the data set (anonymized) to protect the participants' individual privacy. We arranged it so that the clinicians who were responsible for each of the patient cases checked all the filled-in forms. Abuse cases that needed follow-up or report to the police were dealt with by the clinicians responsible.

### Statistical analyses

For statistical analysis, we used IBM SPSS version 29.0 for Windows and Stata version 18. Data on prevalences of the different forms of violence are presented in frequency tables. Gender comparisons of prevalence rates were conducted using Pearson chi-square tests in SPSS, while Truncated Poisson regression for gender comparisons of polyvictimization counts was performed in Stata. Correlations between occurrence rates for the different forms of violence were examined using Spearman correlation test in SPSS. A significance level of 0.05 was used for all statistical tests.

### Results

Table 1 provides selected demographic and socioeconomic characteristics of the families. The majority was from the Middle Hills, including the capital Kathmandu, and most families lived in urban areas. Nuclear family was the most common type of family structure (48%), but extended family structure was also prevalent (43%). About 2% of types of family was labeled "Unknown". These pertained to 16 adolescents who had been referred to the clinic from shelter homes (child protection services) and their original family structure had not been recorded. Parental education varied, with a significant portion having completed secondary education. Most mothers were housewives, while fathers were mostly working in private sector or were government employees. About 13% of the fathers were migrant workers. More than half of the families (54%) had an annual income within the range of 1331–4455 USD.

Table 2 shows the range of occurrence for the different forms of violence in the last 12 months. Most adolescents rated the occurrence of any form of violence in the "Rarely" or "Sometimes" categories (i.e., monthly, or less). Emotional abuse by adults happened more frequently than physical or sexual abuse, with 10% occurrence in the "Frequently" category (i.e., more than once a week to daily). Only 2% of the adolescents reported that sexual abuse happened frequently. Occurrence of neglect and domestic violence was rated as "Frequently" by less than 5% of the adolescents. About 15% of the adolescents reported that emotional types of harassment by peers happened frequently, such as social exclusion, traditional verbal harassment, and online harassment, whereas 6% of the adolescents reported that physical aggression had happened frequently.

Table 3 shows the prevalence of various forms of violence experienced by boys and girls, as well as the overall violence. As seen from the table, 88% of the adolescent patients had experienced some form of violence during the last year. Maltreatment was reported by 84% and peer victimization by 60%. It should be noted, though, that the presented prevalences in this table also include adolescents who reported very low frequency of violence (once or twice occurrence during the last year period). Hence, the prevalences are high. Girls reported higher prevalence of total violence compared to boys. Sexual abuse was more prevalent among girls than boys: 23% versus 8%. Child neglect was reported by 25% of the adolescents and again, was reported more frequently by girls than by boys (30% versus 19%). As for peer victimization, 60% of the adolescents reported that they had experienced it in some form during the last year. Emotional forms of peer victimization were more prevalent. Verbal harassment was reported by 50% of adolescents, whereas physical aggression was reported by 39%. The prevalence of social exclusion for girls was higher among girls: 51%, and for boys, it was 42%.

Table 4 shows that 74% of the adolescents – 71% of the boys and 76% of the girls – had experienced polyvictimization during the last year. The prevalence of the higher numbers (i.e., five and six forms of violence) was 15% for girls and 9%

**Table 1. Distribution of selected socio-demographic variables.**

| Variables | N | % |
|---|---|---|
| *Geographic region* | | |
| Middle Hills (including Kathmandu) | 654 | 80.7 |
| Southern plains/Terai | 134 | 16.5 |
| Himalayan | 22 | 2.7 |
| *Living Area* | | |
| Urban | 517 | 63.8 |
| Semi-urban | 175 | 21.6 |
| Rural | 118 | 14.6 |
| *Types of Family* | | |
| Nuclear family | 387 | 47.8 |
| Extended family | 349 | 43.1 |
| Single parenthood | 58 | 7.2 |
| Unknown | 16 | 2.0 |
| *Parental Education** | | |
| Secondary level | 381 | 47.0 |
| University level | 279 | 34.4 |
| Primary level | 108 | 13.3 |
| Illiterate | 19 | 2.3 |
| *Employment status of parents* | | |
| Unemployed mother | 381 | 47.0 |
| Unemployed father | 62 | 7.7 |
| *Type of employment, mother* | | |
| Housewife | 381 | 47.0 |
| Private sector | 158 | 19.5 |
| Farmer | 85 | 10.5 |
| Other | 91 | 11.2 |
| Government employee | 58 | 7.2 |
| Migrant worker | 37 | 4.6 |
| *Type of employment, father* | | |
| Private sector | 232 | 28.6 |
| Other | 185 | 22.8 |
| Government employee | 121 | 14.9 |
| Farmer | 104 | 12.8 |
| Migrant worker | 102 | 12.6 |
| *Annual family income (USD)*** | | |
| $2,226-$3,340 | 172 | 21.2 |
| $1,331-$2,225 | 138 | 17.0 |
| $3,341-$4,455 | 127 | 15.7 |
| $4,455-$8,905 | 115 | 14.2 |
| $444-$1,330 | 90 | 11.1 |
| ≥$8,905 | 85 | 10.5 |
| <$444 | 78 | 9.6 |

*In the households with two parents, the higher education level was used. Primary level education consists of 5 years of education from grade 1–5. Secondary level education consists of 12 years of education up to grade 12. University level education includes 13 years of education and above.

**Classification of annual income was done according to the Modified Kuppuswamy's Socioeconomic Status Scale in the context of Nepal.

**Table 2. Occurrence rates for different types of violence as experienced by the adolescents during the last year.**

| Types of violence | Range of occurrence* | | | | |
|---|---|---|---|---|---|
| | Never N (%) | Rarely N (%) | Sometimes N (%) | Often N (%) | Frequently N (%) |
| Physical abuse | 455 (56.2) | 82 (10.1) | 172 (21.2) | 66 (8.1) | 35 (4.3) |
| Emotional abuse | 207 (25.6) | 243 (30.0) | 184 (22.7) | 96 (11.9) | 80 (9.9) |
| Sexual abuse | 681 (84.1) | 50 (6.2) | 39 (4.8) | 21 (2.6) | 19 (2.3) |
| Child neglect | 608 (75.1) | 42 (5.2) | 101 (12.5) | 40 (4.9) | 19 (2.3) |
| Domestic violence | 486 (60.0) | 66 (8.1) | 179 (22.1) | 42 (5.2) | 37 (4.6) |
| Physical aggression by peers | 494 (61.0) | 44 (5.4) | 143 (17.7) | 81 (10.0) | 48 (5.9) |
| Verbal harassment by peers | 408 (50.4) | 66 (8.1) | 197 (24.3) | 79 (9.8) | 60 (7.4) |
| Social exclusion by peers | 432 (53.3) | 65 (8.0) | 184 (22.7) | 73 (9.0) | 56 (6.9) |
| Online harassment by peers | 709 (87.5) | 42 (5.2) | 48 (5.9) | 6 (0.7) | 5 (0.6) |

*0 = never, 1 = rarely (once to twice in a year), 2 = sometimes (once in a month or less), 3 = fairly often (1–2 times in a month), or 4 = frequently (a couple of times in a week, or daily).

**Table 3. Prevalence of violence as experienced by boys and girls.**

| Forms of violence | Boys N = 392 (48.4%) | Girls N = 418 (51.6%) | Total N = 810 (100%) | Chi-Square (p-value) |
|---|---|---|---|---|
| **Maltreatment** | 319 (81.4) | 365 (87.3) | 684 (84.4) | 5.4 (0.02) |
| Child abuse | 301 (76.8) | 345 (82.5) | 646 (79.8) | 4.1 (0.04) |
| *Physical abuse* | 186 (47.4) | 169 (40.4) | 355 (43.8) | 4.0 (0.04) |
| *Emotional abuse* | 282 (71.9) | 321 (76.8) | 603 (74.4) | 2.5 (0.11) |
| *Sexual abuse* | 32 (8.2) | 97 (23.2) | 129 (15.9) | 34.1 (<0.01) |
| Child neglect | 75 (19.1) | 127 (30.4) | 202 (24.9) | 13.6 (<0.01) |
| Domestic Violence | 145 (37.0) | 179 (42.0) | 324 (40.0) | 2.8 (0.09) |
| **Peer Victimization** | 227 (57.9) | 261 (62.4) | 488 (60.2) | 1.74 (0.19) |
| *Physical aggression* | 159 (40.0) | 157 (37.0) | 316 (39.0) | 0.77 (0.38) |
| *Verbal harassment* | 197 (50.3) | 205 (49.0) | 402 (49.6) | 0.12 (0.73) |
| *Social exclusion* | 165 (42.1) | 213 (51.0) | 378 (46.7) | 6.39 (0.01) |
| *Online harassment* | 50 (12.2) | 51 (12.8) | 101 (12.5) | 0.06 (0.81) |
| **Total Violence** | 334 (85.2) | 380 (90.9) | 714 (88.1) | 6.30 (0.01) |

**Table 4. Prevalence of polyvictimization*.**

| Forms of polyvictimization | Boys N = 392 (48.4%) | Girls N = 418 (51.6%) | Total N = 810 (100%) |
|---|---|---|---|
| **Zero forms of violence** | 58 (14.8) | 38 (9.1) | 96 (11.9) |
| **One form of violence** | 56 (14.3) | 63 (15.1) | 119 (14.7) |
| **Two forms of violence** | 91 (23.2) | 86 (20.6) | 177 (21.9) |
| **Three forms of violence** | 83 (21.2) | 93 (22.2) | 176 (21.7) |
| **Four forms of violence** | 69 (17.6) | 74 (17.7) | 143 (17.7) |
| **Five forms of violence** | 26 (6.6) | 40 (9.6) | 66 (8.1) |
| **Six forms of violence** | 9 (2.3) | 24 (5.7) | 33 (4.1) |
| **Total polyvictimization** | 278 (70.9) | 317 (75.8) | 595 (73.5) |

*Polyvictimization is defined as experiencing two or more forms of violence. Here, we have examined six different forms: physical violence; emotional violence; sexual abuse; child neglect; domestic violence; peer victimization. Two forms include any two forms of violences, three forms mean any three forms of violence, etc.

for boys. Polyvictimization counts were significantly higher for girls compared to boys, as indicated by the truncated Poisson regression (Wald $\chi^2 = 10.64$, $p < .01$). The model showed that girls had a 17% higher incidence rate ratio (IRR = 1.17) of polyvictimization compared to boys.

Table 5 shows Spearman's correlations between the different forms of violence. Positive correlations were found among all forms of violence; however, the strength of correlations varied. Emotional abuse was strongly correlated with physical abuse and with peer victimization. Domestic violence was moderately correlated with neglect, with physical and emotional abuse, and with peer victimization. Sexual abuse was moderately associated with emotional abuse and with neglect. Low associations were observed between the other forms of violence.

## Discussion

The main aim of this study was to examine last year prevalence and range of occurrence of different types of violence as experienced by adolescent patients attending a child- and adolescent outpatient unit in Nepal. Correlations between the different types of violence were analyzed to explore possible polyvictimization. Most of the participants came from the Middle Hills region of Nepal, including the capital Kathmandu. They were mainly from nuclear families (48%), but nearly as many came from extended families (43%). The majority had a low middle-class background. Most of the parents had a secondary-level education, supporting the World Bank's classification and UNICEF's claim that average education levels are below secondary in Nepal [38].

The study revealed that 88% of the adolescent participants had experienced some form of violence in the last year, and 84% had faced maltreatment. The total prevalence was higher than the global prevalence of violence against children reported in an international review study estimated at 76% [24]. This may be due to the difference in the age range as our study included an older age group (adolescents aged 12–15-years), whereas the global prevalence estimate was based on a broader age range (2–17 years). Further, our study was a clinical sample study, whereas the review study used general population data which likely resulted in a lower prevalence. The higher prevalence in our study may also be because it was done in a LMIC with a high degree of poverty, negative social- and gender norms and inequalities which create an environment in which violence against both girls and boys is more likely to happen [26]. Our findings align with previous research indicating that LMICs exhibit higher prevalence rates of child violence, including polyvictimization, compared to HICs, such as the wealthier European countries [24,39,40]. Further, the high prevalence of child violence in our study might be due to cultural factors such as the traditional use of physical punishment as part of disciplinary child-rearing practices in Nepal [41]. The fact that 88% of adolescent patients reported experiencing some form of violence is highly significant. The finding suggests that a substantial proportion of these adolescents may be seeking psychiatric care due to the psychological consequences of violence exposure. Although this relationship was not directly examined in the current study, it is important to be aware of the potential psychiatric symptoms and mental health outcomes frequently associated with experiences of violence in adolescence [2,11–15]. More awareness and competence among clinicians working

**Table 5. Correlations between the different forms of violence.**

| Domains | PA | EA | SA | N | DV | PV |
|---|---|---|---|---|---|---|
| PA | 1.0 | 0.49** | 0.18** | 0.14** | 0.25** | 0.37** |
| EA | 0.49** | 1.0 | 0.25** | 0.22** | 0.36** | 0.50** |
| SA | 0.18** | 0.25** | 1.0 | 0.26** | 0.24** | 0.15** |
| N | 0.14** | 0.22** | 0.20** | 1.0 | 0.38** | 0.08* |
| DV | 0.25** | 0.36** | 0.24** | 0.38** | 1.0 | 0.28** |
| PV | 0.37** | 0.50** | 0.15** | 0.08* | 0.28** | 1.0 |

Note: *p < 0.05; **p < 0.005.

Physical abuse (PA), Emotional abuse (EA), Sexual abuse (SA), Neglect (N), Domestic violence (DV), Peer victimization (PV).

in child- and adolescent psychiatric units about child violence is crucial. A recent systematic international review and meta-analysis reported that about one in two health professionals face situations of violence against children and adolescents in their clinical practice (41%), and that approximately only one in three health professionals report the cases (30%) [42].

Due to the scarce amount of internationally published Nepali research studies on child violence, comparisons with other studies from Nepal are limited. However, a recent general population study on child maltreatment (i.e., child abuse, child neglect, and domestic violence) should be mentioned. This is a study among school-going adolescents who were recruited from 20 public schools in Kathmandu [43]. It showed a self-reported past-year prevalence of child maltreatment at 88%, higher than the prevalence in our study (84%). The higher prevalence may be explained by sample characteristics. Many of the students came from poor, low-income families as children from low-income families generally receive education from the free-to-attend public schools [44]. Research has shown that poverty is linked to child violence in several ways, increasing its prevalence [43,45]. Further, the collection of data in the study was carried out four months after the Nepal earthquake of 25 April 2015, and natural disasters are associated with an increased risk of child violence [46].

However, there might be other, unknown explanations as well. We should bear in mind that differences in prevalence across studies may be attributed to many factors, like differences in the operational definition of the various forms of child violence, differences in study design, age groups, data collection methods, differences in the instruments used, and variations in the specific populations examined.

The present study found high prevalence rates for the different types of maltreatment: physical abuse at 44%, emotional abuse at 74%, sexual abuse at 16%, neglect at 25%, and domestic violence at 40%. However, it should be noted that the prevalences presented in our study were based on a broad range of occurrence, i.e., from rarely (once to twice in a year) to frequently (a couple of times in a week or daily) which might be a broader range than that used in some other studies. When comparing the occurrence of abuse for the "Frequently" category only (i.e., weekly to daily abuse) with the "Severe/Very Severe" category in a clinical study from another LMIC, Iran, we found higher prevalence in our study, particularly for emotional abuse. In the Iranian study, which used a sample of children 8–18 years who were referred to psychiatric out-patient facilities, the summed-up prevalence rates of severe and very severe emotional abuse, physical abuse, and sexual abuse were found to be 3.8%, 3.8%, and 0.0%, respectively [47] whereas in our study, the occurrence rates for the "Frequently" category were 9.9%, 4.3%, and 2.3%, respectively. However, frequency and severity are only partly overlapping concepts in this respect. Besides, the age groups in the two studies differed. Hence, comparisons should be interpreted with caution.

Our study indicates higher rates of child abuse and neglect in clinical psychiatric samples in Nepal compared to clinical studies in high-income countries (HICs) like Norway and the USA [48–50]. An earlier study from Norway that examined a national sample of adolescents admitted to child- and adolescent psychiatric outpatient clinics, found that 60.2% of the adolescents had been abused or neglected [49]. This prevalence was lower than the prevalence of abuse in our sample (79.8%). Also, our study found a higher prevalence compared to an earlier study from the USA that found a lifetime incidence of 30% of physical and sexual abuse in a clinical sample of children admitted to psychiatric units [50]. The prevalence of maltreatment (84%) found in our study was again higher than in a more recent US-based clinical study among adult patients in outpatient psychiatric services which found that 3 out of 4 patients had experienced childhood maltreatment [51]. It should be noted, though, that the findings might not be directly comparable due to possible recall bias in the latter study that was based on adult reports.

## Emotional abuse

Emotional abuse was the most frequently reported form of abuse, with an overall prevalence of 74.4%, slightly higher for girls than for boys (76.8% and 71.9%, respectively). About 22% of the adolescents reported that emotional types of assault happened often or frequently. The overall prevalence of emotional abuse was like reported by the Nepal Multiple

Indicator Cluster Survey (the 2014 NMICS), which showed that 77.3% of the children included in the study had experienced emotional abuse [52,53]. The prevalence found for emotional abuse in our study also align with the previously mentioned Nepali school study which reported a prevalence of emotional abuse of 75.2% [43].

According to a UNICEF report from 2020, emotional abuse was found to be prevalent throughout the South Asian region and almost always present when other types of violence are measured, suggesting that all forms of violence against children contain elements of emotional violence within the region [26]. The high rate of emotional abuse is probably influenced by harsh disciplinary practices that are still used both in schools and homes in Nepal. Additionally, the absence of the head of the family due to immigrant work can lead to increased use of harsh and critical comments to exert power and control over adolescents [41]. In our study, 17.2% of the parents were migrant workers.

The overall prevalence of emotional abuse in our study was higher that the prevalence that has been reported for emotional abuse globally. Further, we found rather small gender differences, lying within the range of 71–77% for both boys and girls. These findings align with the results from an international review study which reported smaller gender differences in Asia compared to North America and Europe [54].

## Physical abuse

Even though Nepal's law has forbidden any form of child violence since 2018, the present study found a high rate of physical abuse (prevalence of 43.8%) which may reflect inadequate knowledge in the Nepali population of children's needs and a lack of awareness of the negative impact of physical disciplinary methods. Our finding is in line with a population-based, cross-sectional Nepali study from 2017 which found a similarly high prevalence of physical abuse, suggesting that corporal punishment of children was common across Nepal [26,41]. It is also in line with findings from the Nepal Multiple Indicator Cluster Survey which found that 49.8% of the children who participated had experienced moderate physical abuse and 21.5% had experienced more severe forms [53].

The high prevalence of physical abuse in our study also supports previous data from cross-sectional studies in the South Asian region showing that approximately half of all children had experienced some type of physical violence during their lifetime [26]. Corporal punishment of children is prevalent throughout the South Asian region and might take place both in the homes and in the school context. For instance, nationally representative data in neighboring country Bhutan found that the most common forms of physical violence against children were committed in the context of corporal punishment in schools [26]. In Nepal, future studies examining corporal punishment of children in schools are warranted.

In the present study, the prevalence for boys and girls differed, boys experiencing more physical abuse (47.4%) than girls (40.4%). However, this is a rather small difference. Our finding aligns with a review study which demonstrated that gender difference in physical abuse varies across countries and continents and might be rather small in Asia as compared to Europe. In European studies, physical abuse was found to be much higher for boys (27.0%) than for girls (12.0%) [54]. However, more studies in Asian countries, including Nepal, are needed to verify the smaller gender difference and to explore possible cultural- and contextual determinants.

## Sexual abuse

The prevalence of sexual abuse in our study was 15.9%. This was somewhat higher than last-year prevalence of 11.3% that was reported by a school-study in the general Nepali population [42]. It was also higher than the general population estimate of sexual child abuse for the South Asian region which was reported to be 9–14% [26]. This finding is not surprising as sexual abuse may be more prevalent in clinical samples of patients seeking help in the mental health services.

We found that sexual abuse was more common among the girls in our sample, affecting almost one in four, whereas less than one in ten boys reported such past-year experiences. The finding is consistent with an international review that reported a generally higher prevalence for girls than for boys globally, although prevalences tended to vary considerably across continents. According to the review, prevalence estimates for Asia of self-reported sexual abuse were 9.0% for girls

and 6.7% for boys [54]. The higher prevalence for girls may be due to negative gender norms and inequalities that exists in many countries, including Nepal.

A similar gender difference for sexual abuse (i.e., higher prevalence for girls) has been demonstrated in clinical samples as in general population samples [51,31]. Our study confirms this finding. However, more extensive studies on sexual abuse of girls are warranted in Nepal, including studies of child marriage which has a high prevalence in the country [55].

## Child neglect and domestic violence

The prevalence of child neglect in our study was 24.9%, and it was higher for girls (30.4%) than for boys (19.1%). About 5% of the adolescents reported that abuse occurred often (1–2 times a month), and 2.3% reported that it occurred frequently (a couple of times a week or daily). This prevalence was higher than that found in a former meta-analytic review which had an estimated global prevalence of self-reported physical neglect of 16.3%, and a prevalence of emotional neglect of 18.4% [25]. Our finding is consistent with findings reported in a more recent research paper based on published reviews and meta-analyses in the field of child neglect, which showed a global prevalence between 16 and 26%, and with a higher prevalence in LMICs, in clinical groups, and in self-reported studies [56]. However, due to the dearth of studies on child neglect in LMICs, estimates from these countries are less valid which makes prevalences of child neglect between LMICs and HICs difficult to compare.

The higher prevalence of neglect for girls in our study was consistent with a Nepali study of school-going adolescents which reported that female students were more likely to report neglect than boys [48]. However, it was in odds with data from a systematic international review which found approximately similar prevalence estimates for girls and boys on the Asian continent (girls: 26.3%; boys: 23.8%) [54]. More studies on child neglect in the South Asian countries, including Nepal, are warranted to establish reliable and valid estimates of prevalence of child neglect in this region and to compare with findings in other countries.

Domestic violence was experienced by 40% of the adolescents. The high prevalence confirms the findings by other studies in the South Asian region which showed that witnessing violence, either between parents or between known adults, is frequently reported by children [23]. Our finding is also in line with the generally higher prevalence in LMICs that was reported in a recent global review of childhood exposure to physical domestic and family violence [57]. However, domestic violence in our study exceeded that which was reported for South Asia and America in the same review, estimated at 32.5% for victims and 29.1% for witnesses [57]. The higher prevalence estimates of domestic violence in LMICs, like Nepal, may reflect underlying hash economic conditions, patriarchal cultural norms and traditions [58] and lower service availability. The higher prevalence in the present study might also be due to the use of a clinical sample.

Further, our study found that the exposure of domestic violence reported by boys and girls was approximately similar (girls: 42%; boys: 37%). The finding underpins findings reported internationally, suggesting that witnessing physical domestic and family violence did not differ between boys and girls [57].

## Peer victimization

We found that 60.2% of the adolescents had experienced some form of peer victimization in the past year (boys: 62.4%; girls: 60.2%), with emotional victimization being the most common form. This prevalence was somewhat higher than in a recent school-based Nepali study in the general population which found an overall prevalence of peer victimization in adolescents of 50.7% (boys: 55.7%; girls: 46.2%) [59]. The prevalence in our study was much higher compared to the overall prevalence of 34.4% reported in a larger survey done in 68 LMICs, using data from the same Global School-based Student Health Survey (GSHS) as the Nepali study [60]. Further, our findings support previous findings reported in a systematic review for the South Asian countries which showed that school violence and bullying was prevalent in this region [26]. However, our findings of gender differences were in odds with the data for the age group 13–15 years for the region. Whereas our study found similar prevalence of peer victimization for boys and girls, the South Asian review study reported

more overall bullying, and more physical bullying in boys than in girls [26]. Our finding of more similar prevalences might be explained by a possible trend in the region of increasing physical victimization among girls and decreasing physical victimization among boys [26]. However, more Nepali studies, including studies that measure peer victimization data over time, are warranted to gain more valid estimates and better understanding of the phenomenon.

Finally, the prevalence of peer victimization found in our study was much higher than the mean of the prevalence of 28,9% for peer victimization which was reported in an international study of school going adolescents in 13 European and Asian countries [61]. However, the rates of prevalence varied widely between countries, suggesting that cultural factors may play a role in peer victimization. The higher prevalence rate of peer victimization for Nepal might be due various reasons, e.g., social acceptance of corporal punishment as a disciplinary method, higher acceptance of gender-based violence, traditional caste-based social inequalities, and lack of awareness or ignorance concerning the potentially serious consequences that peer harassment might have on child mental health.

The prevalence of online harassment in our study was 12.5%, compared to 5.1% for cybervictimization only and 6.1% for combined victimization (traditional and cybervictimization) in the Eurasian study [61]. Online harassment or cyberbullying is largely unexplored in LMICs like Nepal and should be more focused in future research.

## Polyvictimization

Global data highlights that nearly three quarters of children experience at least one of the many forms of violence1. New global evidence highlights the magnitude of the polyvictimization of children, or children experiencing multiple forms of violence or violence in multiple settings, however, this area is still under-researched in the region.

We found that 73.5% of the adolescents had experienced multiple forms of violence in the past year. Girls had a higher incidence rate ratio of polyvictimization compared to boys. Our finding is consistent with a survey among public secondary school students in Kathmandu, which found a similarly high proportion of polyvictimization (78%) among adolescents in the previous year [43]. A recent review and meta-analysis on polyvictimization among children and adolescents in LMICs showed that prevalence ranged from 0.3% to 74.7% with an overall estimate of 38.1%, and that experiences of polyvictimization were more prevalent than in HICs [27]. However, the area of polyvictimization is still under-researched in the South Asian region and few studies on child violence have included it [26]. Acknowledging that the context-specific nature of polyvictimization is important, more such studies are warranted in Nepal, especially studies that explore possible gender differences [28]. Also, studies of polyvictimization in families with migrant parents are recommended due to the high proportion of migrants in the country. A recent Chinese study showed that left-behind children were exposed to a higher level of polyvictimization than children living with both non-migrant parents in rural China [62].

Comparing our results with other Asian studies, we found that the prevalence in our study was higher than in studies from some other parts of Asia, e.g., in a large Chinese study which reported a life-time prevalence of 14% among adolescents aged 15–17 years in the general population [63]. A study from the USA that collected data across 56 centers providing youth mental health services, revealed that about half of the youth aged 13–18 years experienced polyvictimization [64], which is lower than the prevalence found in the present study.

The various prevalence estimates between countries suggest that polyvictimization may be influenced by cultural, social, and environmental factors. In addition, the inconsistency in methods of defining and measuring polyvictimization makes comparisons difficult, and there is a need to establish a valid construct that is consistently agreed upon in the research community [28,64].

## Strengths and limitations

Major strengths of the study include a substantial clinical sample size of 810 adolescents and a nearly equal gender distribution, which adds robustness. It is a rigorous study, providing detailed data on various forms of violence, including physical, emotional, and sexual abuse, neglect, domestic violence, peer aggression, and polyvictimization. The age range

of the participants, i.e., 11–15 years, represents a critical developmental stage in life, making the findings particularly relevant. Our study was based on adolescent self-reports of last year experiences which implied less risk of memory bias compared to childhood studies using adult informants or life-long experiences. The screening forms that were used for assessing maltreatment, were taken from validated screening instruments that had been used in other studies internationally. The participation rate was 85% which must be regarded as high. However, the study has limitations. There is a potential for reporting bias as adolescents might underreport or overreport their experiences due to social stigma, fear, or memory recall issues. The data was collected for one hospital only, and although the intake of patients was from the whole country, it limits the generalizability of the study. Further, our study is a survey study with less scrutiny as to assessment. More extensive, validated multi-item questionnaires and corroborating reports from multiple respondents might have strengthened its validity. Also, the use of additional qualitative methods and a more in-depth, intervention-focused approach might have provided a deeper understanding of the phenomenon of child violence.

Unfortunately, the more underprivileged castes and ethnic groups were underrepresented in our sample, possibly caused by less access of services and more illiterate populations. Future cross-country studies are warranted to identify specific types of abuse and neglect in particular cultures and settings.

Further, assessing social- and family correlates was not within the scope of the current study and should be addressed in future studies. Also, assessing long-term mental health consequences of the different forms of violence was not part of this study. It is strongly recommended that future Nepali studies use a longitudinal design to assess the long-term psychological impact.

Finally, the comparisons with other studies can be questioned as to their validity as many studies were not identical as to the definition of concepts of child violence, sample characteristics, and methods. Comparisons with global estimates of child violence were particularly difficult as in these reviews and meta-analyses, there was an overrepresentation of studies from high-income countries with predominantly Caucasian populations, particularly from the U.S.A. Therefore, our comparisons with global prevalence estimates may be biased in favor of this demographic.

## Conclusions

The high total prevalence of the different forms of violence among adolescents attending a child and adolescent psychiatric outpatient clinic in Nepal, highlights a serious public health- and human rights issue in the country. The finding reinforces the urgent need for integrated mental health and psychosocial interventions for this vulnerable population. The study is the first study in Nepal to measure the prevalence and the range of occurrence of the different forms of child violence in a clinical sample of adolescent psychiatric patients and adds to the sparse literature on child violence in LMICs. Our findings highlight an overall high prevalence of child violence compared to other international studies. Prevalence rates were high for both boys and girls, girls experiencing similar or slightly more violence than boys. However, the experience of neglect and sexual abuse was much higher for girls than for boys. The correlations that were found between different forms of violence, together with the high prevalence of polyvictimization, suggest a compounded risk of mental health problems, indicating a need for holistic intervention strategies. The limitations of the study suggest that more research in the field of child violence in Nepal is warranted to produce more reliable prevalence estimates for the country that may guide policy and intervention efforts.

## Supporting information

**S1 File. Paper I data Plos one.**
(SAV)

**S2 File. PLOS ONE global inclusivity.**
(DOCX)

## Acknowledgments

We extend our heartfelt appreciation to our professors, teachers and experts in the field of child and adolescent psychiatry for their invaluable guidance and support throughout the course of this study. We are deeply grateful to all the participants and their parents for their time, cooperation, and willingness to contribute. We also wish to express our sincere gratitude to the CWIN-Nepal and the Child and Adolescent Psychiatry Unit at Kanti Children's Hospital for their generous support and collaboration, which played a crucial role in the successful completion of this research.

## Author contributions

**Conceptualization:** Rampukar Sah, Per Håkan Brøndbo, Anne Cecilie Javo.

**Data curation:** Rampukar Sah, Jasmine Ma, Bjørn Helge Handegård, Anne Cecilie Javo.

**Formal analysis:** Rampukar Sah, Per Håkan Brøndbo.

**Methodology:** Rampukar Sah, Per Håkan Brøndbo, Jasmine Ma, Ketil Lenert Hansen, Narmada Devkota, Anne Cecilie Javo.

**Project administration:** Rampukar Sah, Per Håkan Brøndbo, Anne Cecilie Javo.

**Resources:** Rampukar Sah.

**Supervision:** Per Håkan Brøndbo, Ketil Lenert Hansen, Narmada Devkota, Bjørn Helge Handegård, Anne Cecilie Javo.

**Validation:** Bjørn Helge Handegård.

**Visualization:** Rampukar Sah.

**Writing – original draft:** Rampukar Sah.

**Writing – review & editing:** Rampukar Sah, Per Håkan Brøndbo, Jasmine Ma, Ketil Lenert Hansen, Narmada Devkota, Bjørn Helge Handegård, Anne Cecilie Javo.

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
