## [Decision Letter · Decision Letter 0]

25 Jul 2025

Dear Dr. Sah,

Thank you for submitting your manuscript to PLOS ONE. After careful consideration, we feel that it has merit but does not fully meet PLOS ONE’s publication criteria as it currently stands. Therefore, we invite you to submit a revised version of the manuscript that addresses the points raised during the review process.

We look forward to receiving your revised manuscript.

Kind regards,

Shivanand Kattimani

Academic Editor

PLOS ONE

Journal Requirements:

3. In the online submission form you indicate that your data is not available for proprietary reasons and have provided a contact point for accessing this data. Please note that your current contact point is a co-author on this manuscript. According to our Data Policy, the contact point must not be an author on the manuscript and must be an institutional contact, ideally not an individual. Please revise your data statement to a non-author institutional point of contact, such as a data access or ethics committee, and send this to us via return email. Please also include contact information for the third party organization, and please include the full citation of where the data can be found.

Reviewers' comments:

Reviewer's Responses to Questions

**Comments to the Author**

1. Is the manuscript technically sound, and do the data support the conclusions?

Reviewer #1: Yes

Reviewer #2: Partly

2. Has the statistical analysis been performed appropriately and rigorously?

Reviewer #1: Yes

Reviewer #2: No

3. Have the authors made all data underlying the findings in their manuscript fully available?

Reviewer #1: Yes

Reviewer #2: No

4. Is the manuscript presented in an intelligible fashion and written in standard English?

Reviewer #1: Yes

Reviewer #2: Yes

Reviewer #1: Introduction

The introduction is well-founded, supported by global data that effectively justifies the need for this study. Excellent work.

Methods and Materials

• Under “Study Site and Population”, include the following statement regarding ethical approval:

“This study was approved by the Kanti Children's Hospital Ethical Review Committee (Reference No: 621) and the Nepal Health Research Council (Reference No: 1227).”

• In the “Procedure” section, remove the following sentence from the end of the paragraph:

“All participating adolescents received a small gift as a token of their help.”

Results

• In Table 1 – Distribution of Selected Socio-Demographic Variables, organize the data in descending order based on the absolute number (N). See the example below for formatting guidance:

Variables N %

Middle Hills (including Kathmandu) 654 80.7

Southern Plains/Terai 134 16.5

Himalayan 22 2.7

Discussion

Enhance the discussion related to the prevalence of violence in a clinical sample of adolescent patients attending a child and adolescent psychiatric outpatient clinic. The fact that 88% of adolescents reported experiencing some form of violence is highly significant. This finding suggests that a substantial proportion of these adolescents may be seeking psychiatric care due to the psychological consequences of violence exposure. Although this relationship was not directly examined in the current study, it is important to highlight this point in the discussion section. Doing so would raise awareness among readers about the potential psychiatric symptoms and mental health outcomes frequently associated with experiences of violence in adolescence.

Conclusions

Clearly emphasize the prevalence of violence in a clinical sample of adolescent patients attending a child and adolescent psychiatric outpatient clinic in Nepal, underlining that 88% of participants had experienced some form of violence. This finding reinforces the urgent need for integrated mental health and psychosocial interventions for this vulnerable population.

Reviewer #2: Thank you for the opportunity to review this manuscript examining the prevalence, range, and interrelationships of various forms of violence among adolescents in Nepal. The topic is timely, sensitive, and of public health relevance, particularly in low- and middle-income countries. The large sample size and inclusion of different types of violence (including peer-related victimization and polyvictimization) add value to the manuscript. However, I have several methodological and reporting concerns that must be addressed before this manuscript can be considered for publication.

The authors rely heavily on descriptive statistics and bivariate Spearman correlations. While these provide foundational information, there is an absence of more rigorous analyses that could better capture the complexity of the data.

Recommendation: Consider conducting multivariate regression models (e.g., logistic regression for each form of violence by gender, socioeconomic variables, etc.) or latent class analysis to identify patterns of victimization.

Additionally, testing for gender differences (e.g., chi-square tests or logistic regressions with interaction terms) would offer more robust support for claims of gender disparities.

Summary

While the topic and data are of clear value, the manuscript in its current form falls short of PLOS ONE’s standards for methodological robustness and transparency. I encourage the authors to consider more advanced analyses, justify statistical decisions more clearly, and improve reporting clarity. If these revisions are addressed, I would be inclined to support publication.

Recommendation: Major revision

**Do you want your identity to be public for this peer review?** For information about this choice, including consent withdrawal, please see our Privacy Policy

Reviewer #1: **Yes: ** LUCAS JAMPERSA

Reviewer #2: **Yes: ** ERNESTINA AIDOO

---

## [Author Response · Author response to Decision Letter 1]

15 Sep 2025

Response to the Academic Editor and Reviewers

Manuscript ID: PONE-D-25-20724

Title: Prevalence of violence in a clinical sample of adolescent patients visiting a child and adolescent psychiatric outpatient clinic in Nepal

Journal: PLOS ONE

Dear Academic Editor and Reviewers,

We sincerely appreciate the time and effort you have dedicated to reviewing our manuscript and providing constructive feedback. We have carefully considered all comments and have revised the manuscript accordingly. Below, we provide a point-by-point response to each concern raised.

A. The requirements made by the Academic Editor

Requirement 1: “Please ensure that your manuscript meets PLOS ONE's style requirements, including those for file naming”.

Our response:

We have now done as required to the best of our abilities.

Requirement 2: “Please include a complete copy of PLOS’ questionnaire on inclusivity in global research in your revised manuscript. Our policy for research in this area aims to improve transparency in the reporting of research performed outside of researchers’ own country or community. The policy applies to researchers who have travelled to a different country to conduct research, research with Indigenous populations or their lands, and research on cultural artefacts”.

Our response:

We have filled in the PLOS’ questionnaire on inclusivity in global research. However, it seems this policy applies to researchers who have travelled to a different country to conduct research, and we would like to inform you that this does not apply to our study. The research has not been performed outside of Nepal which is the main researcher’s / first authors’ own country. The first author is a PhD student attached to the UiT – The Arctic University of Norway – but he has not gone abroad to study and is working the whole time from Nepal. The third and the fifth authors of this paper are also Nepali researchers (and with a PhD) who are living and working in their own country, Nepal. Additionally, this research was conducted at a tertiary hospital in Kathmandu using a clinical sample, not in any of the local communities of Nepal.

Requirement 3: “In the online submission form, you indicate that your data is not available for proprietary reasons and have provided a contact point for accessing this data. Please note that your current contact point is a co-author on this manuscript….”

Our response:

We are sorry – this was a mistake, All raw data will be made fully available without restriction to the PLOS ONE as part of supportive information. The Data Availability Statement in the manuscript will be updated accordingly.

Requirement 4: “We note that the grant information you provided in the ‘Funding Information’ and ‘Financial Disclosure’ sections do not match”.

Our response:

Sorry – we were not aware. We have now corrected this so that identical information is given.

Requirement 5: “If the reviewer comments include a recommendation to cite specific previously published works, please review and evaluate these publications to determine whether they are relevant and should be cited. There is no requirement to cite these works unless the editor has indicated otherwise”.

Our response:

We have extended the Discussion part according to the comments made by Reviewer. Included in this extension, we have added one additional, relevant study as a reference – see our response to Reviewer 1, Comment 5 below.

B. Comments from the reviewers

Reviewer #1:

Reviewer 1, Comment 1: “Introduction: The introduction is well-founded, supported by global data that effectively justifies the need for this study. Excellent work”.

Our response:

We thank Reviewer #1 for his positive feedback.

Reviewer 1, Comment 2: “Methods and Materials: Under “Study Site and Population”, include the following statement regarding ethical approval: “This study was approved by the Kanti Children's Hospital Ethical Review Committee (Reference No: 621) and the Nepal Health Research Council (Reference No: 1227)”.

Our Response:

We have added the requested statement under "Study Site and Population": "This study was approved by the Kanti Children's Hospital Ethical Review Committee (Reference No: 621) and the Nepal Health Research Council (Reference No: 1227)." - see page 7 and page 8, in the revised manuscript.

Reviewer 1, Comment 3: “In the “Procedure” section, remove the following sentence from the end of the paragraph: “All participating adolescents received a small gift as a token of their help.”

Our Response:

As recommended, we have removed the sentence: "All participating adolescents received a small gift as a token of their help." - see page 8, in the revised manuscript.

Reviewer 1, Comment 4: “Results: In Table 1 – Distribution of Selected Socio-Demographic Variables, organize the data in descending order based on the absolute number (N). See the example below for formatting guidance:

Variables N %

Middle Hills (including Kathmandu) 654 80.7

Southern Plains/Terai 134 16.5

Himalayan 22 2.7”

Our Response:

Thank you for your advice, which makes the table easier to view. We have now reorganized Table 1 in descending order based on absolute numbers (N) as recommended – see page 12-`14 in the revised manuscript

Reviewer 1, Comment 5: “Discussion: Enhance the discussion related to the prevalence of violence in a clinical sample of adolescent patients attending a child and adolescent psychiatric outpatient clinic. The fact that 88% of adolescents reported experiencing some form of violence is highly significant. This finding suggests that a substantial proportion of these adolescents may be seeking psychiatric care due to the psychological consequences of violence exposure. Although this relationship was not directly examined in the current study, it is important to highlight this point in the discussion section. Doing so would raise awareness among readers about the potential psychiatric symptoms and mental health outcomes frequently associated with experiences of violence in adolescence”.

Our Response:

Thank you for this important advice. We agree that this aspect was lacking in the discussion. We have now expanded the discussion to explicitly highlight the high prevalence (88%) of violence exposure in this clinical sample and its potential implications for psychiatric symptoms. A new paragraph has been added, see page 21 in the revised manuscript which now reads as follows:

“The fact that 88% of the adolescent patients reported experiencing some form of violence is highly significant. This finding suggests that a substantial proportion of these adolescents may be seeking psychiatric care due to the psychological consequences of violence exposure. Although this relationship was not directly examined in the current study, it is important to be aware of the potential psychiatric symptoms and mental health outcomes frequently associated with experiences of violence in adolescence2,11-15. More awareness and competence among clinicians working in child- and adolescent psychiatric units about child violence is crucial. A recent systematic review and meta-analysis reported that about one in two health professionals face situations of violence against children and adolescents in their clinical practice (41%), and that approximately only one in three health professionals report the cases (30%)42.”

Reviewer 1, Comment 6: In the “Conclusions”, clearly emphasize the prevalence of violence in a clinical sample of adolescent patients attending a child and adolescent psychiatric outpatient clinic in Nepal, underlining that 88% of participants had experienced some form of violence. This finding reinforces the urgent need for integrated mental health and psychosocial interventions for this vulnerable population.

Our Response:

We have revised the conclusion part by adding the following paragraph, see page 31, in the revised manuscript:

"The high total prevalence of different forms of violence (88%) among adolescents attending a child and adolescent psychiatric outpatient clinic in Nepal, highlights a serious public health- and human rights issue in the country. The finding reinforces the urgent need for integrated mental health and psychosocial interventions for this vulnerable population”.

Reviewer #2

Reviewer 2, Comment 1: The topic is timely, sensitive, and of public health relevance, particularly in low- and middle-income countries. The large sample size and inclusion of different types of violence (including peer-related victimization and polyvictimization) add value to the manuscript.

Our Response:

Thank you - we appreciate your approval stated above.

Reviewer 2, Comment 2: The authors rely heavily on descriptive statistics and bivariate Spearman correlations. While these provide foundational information, there is an absence of more rigorous analyses that could better capture the complexity of the data. Recommendation: Consider conducting multivariate regression models (e.g., logistic regression for each form of violence by gender, socioeconomic variables, etc.) or latent class analysis to identify patterns of victimization.

Our response:

Thank you for this comment and giving us the opportunity to explain. Our study is primarily descriptive, focusing on the prevalence and occurrence rates of different forms of violence and their correlations. This foundational work provides a necessary baseline for understanding the extent of violence among adolescents using child- and adolescent psychiatric services in Nepal. We agree that including multivariable regression analyses would have shifted the focus of the paper from descriptive prevalences to explanatory modeling. However, that was beyond the scope of the current aims. We appreciate your suggestion of using latent class analysis (LCA) to identify patterns of victimization. While LCA is a valuable tool, it is beyond the scope of the current study which focuses on the prevalence, gender differences, and correlations between different forms of violence. Also, as the current study does not capture all aspects of violence experience (e.g., severity), it limits the ability to create meaningful latent classes.

Reviewer 2, Comment 3: Additionally, testing for gender differences (e.g., chi-square tests or logistic regressions with interaction terms) would offer more robust support for claims of gender disparities.

Our response:

Gender is a central variable in the study, as one of the stated aims is to assess gender differences in the prevalence of violence. By focusing on gender, the study provides meaningful insights into disparities between boys and girls, which is a critical issue in the context of violence research. We have now added the results from Chi-square tests in Table 3, see page 16-17, in the revised manuscript.

In Table 4 (page 18), we have now included all categories (from zero to six forms of violence), which will give a more comprehensive view of the data and allow readers to contextualize the findings. When including the "Zero forms of violence" and "One form of violence" categories in the table, we provide a complete distribution of the number of violence experience types for boys and girls. This will allow readers to see the full spectrum of victimization type numbers, from no violence to extreme polyvictimization, and better understand the prevalence of violence type numbers in the sample.

Additionally, we performed an additional test: Truncated Poisson regression of polyvictimization by gender - with gender interaction terms - to make a statistical test of gender differences in the distribution of the number of violence forms. The coefficients table from the truncated Poisson regression analysis showed:

PolyVictimization | Coefficient Std. err. z P>|z| [95% conf. interval]

Gender |

Female | .1531651 .04695 3.26 0.001 .0611294 .2452009

_cons | .910035 .03428 26.54 0.000 .8428336 .9772365

The coefficient 0.1531651 for the Female variable is positive, indicating that the polyvictimization counts are higher for girls compared to boys.

We transformed the coefficient (0.1531651) into an incident rate ratio: IRR = exp (0.1531651) = 1.166 ≈ 1.17, meaning that girls had a 17% higher rate of polyvictimization compared to boys

We added this information in the text below Table 4 providing the test results in parenthesis – see page 18-19. We also revised the “Statistic analyses” paragraph in the “Methods” part, informing that we used the Truncated Poisson regression – see page 12. The Truncated Poisson regression analysis was reasonable to use since the dependent variable is a truncated count variable, and the mean and variance for this count variable were on a similar level. This similarity was observed for both boys and girls

In Table 5, we have used the Spearman correlations test to show the correlations between occurrence rates for the different forms of violence. Spearman correlations were employed instead of Pearson correlations due to non-normal distributions and ordinal response scales for violence frequency. We have stated our use of Spearman correlations in the Methods section, Statistical analyses – see page 12 in the revised manuscript.

Finally, in the “Statistic analyses” paragraph in the “Methods” part, we also added information about the p-level used in this paper (page 12 in the revised manuscript) as we had forgotten to mention this.

We believe the revisions described have addressed the concerns raised by the reviewers and that they have strengthened the manuscript. We are grateful for the reviewers’ insights and hope the revised version meets PLOS ONE’s standards.

Sincerely,

Rampukar Sah

UiT-The Arctic University of Norway / CWIN-Nepal CAPMH Research and Outreach Center, Lincoln Marg, Kathmandu 44600, Nepal

rampukar2950@gmail.com

Attachments:

- Revised Manuscript (with tracked changes)

- Clean Manuscript

- Raw data will be made available to PLOS ONE

---

## [Decision Letter · Decision Letter 1]

13 Oct 2025

Prevalence of violence in a clinical sample of adolescent patients visiting a child and adolescent psychiatric outpatient clinic in Nepal

PONE-D-25-20724R1

Dear Dr. Sah,

We’re pleased to inform you that your manuscript has been judged scientifically suitable for publication and will be formally accepted for publication once it meets all outstanding technical requirements.

Kind regards,

Shivanand Kattimani

Academic Editor

PLOS ONE

Additional Editor Comments (optional):

However, please pay attention to comment from one reviwer (Reviwer no3),

"I believe the authors have adequately responded to prior reviewers' comments. My only comment is a minor typographical one: I think the column and row labeled "B" in table 5 should instead be labeled "PV" for peer victimization. If this is in fact a typo and is corrected, then the manuscript would need no further review for me and I would recommend it be accepted for publication."

Reviewers' comments:

Reviewer's Responses to Questions

**Comments to the Author**

Reviewer #1: All comments have been addressed

Reviewer #3: (No Response)

2. Is the manuscript technically sound, and do the data support the conclusions?

Reviewer #1: Yes

Reviewer #3: Yes

3. Has the statistical analysis been performed appropriately and rigorously?

Reviewer #1: Yes

Reviewer #3: Yes

4. Have the authors made all data underlying the findings in their manuscript fully available?

Reviewer #1: Yes

Reviewer #3: Yes

5. Is the manuscript presented in an intelligible fashion and written in standard English?

Reviewer #1: Yes

Reviewer #3: Yes

Reviewer #1: Congratulations on your work. All your requests were met to improve the visualization of your research. I wish you success in publishing.

Reviewer #3: I believe the authors have adequately responded to prior reviewers' comments. My only comment is a minor typographical one: I think the column and row labeled "B" in table 5 should instead be labeled "PV" for peer victimization.

If this is in fact a typo and is corrected, then the manuscript would need no further review for me and I would recommend it be accepted for publication.

**Do you want your identity to be public for this peer review?** For information about this choice, including consent withdrawal, please see our Privacy Policy

Reviewer #1: **Yes: ** LUCAS JAMPERSA

Reviewer #3: No

---

## [Editor Report · Acceptance letter]

PONE-D-25-20724R1

PLOS ONE

Dear Dr. Sah,

I'm pleased to inform you that your manuscript has been deemed suitable for publication in PLOS ONE. Congratulations! Your manuscript is now being handed over to our production team.

Kind regards,

on behalf of

Dr. Shivanand Kattimani

Academic Editor

PLOS ONE